# Advanced and Functional Structured Ceramics: MgF_2_ and ZnS

**DOI:** 10.3390/ma15144780

**Published:** 2022-07-07

**Authors:** Natalia Kamanina, Andrey Toikka, Yaroslav Barnash, Pavel Kuzhakov, Dmitry Kvashnin

**Affiliations:** 1Lab for Photophysics of Media with Nanoobjects, Vavilov State Optical Institute, Kadetskaya Liniya V.O., dom 5, korp.2, 199053 St. Petersburg, Russia; atoikka@obraz.pro (A.T.); barnash.yaroslaw@yandex.ru (Y.B.); kpv_2002@mail.ru (P.K.); 2Photophysica of Nanostructured Materials and Devices, Vavilov State Optical Institute, Babushkina str., dom 36, korp.1, 192171 St. Petersburg, Russia; 3Department of Photonics, St.-Petersburg Electrotechnical University (“LETI”), Ul. Prof. Popova, dom 5, 197376 St. Petersburg, Russia; 4Department of Advanced Development, Petersburg Nuclear Physics Institute, National Research Center “Kurchatov Institute”, 1 md. Orlova Roshcha, 188300 Gatchina, Russia; 5Emanuel Institute of Biochemical Physics RAS, 4 Kosigina st., 119334 Moscow, Russia; dgkvashnin@phystech.edu; 6School of Chemistry and Technology of Polymer Materials, Plekhanov Russian University of Economics, Stremyanny Lane, 36, 117997 Moscow, Russia; 7Department of Physics and Mathematics, Pirogov Russian National Research Medical University, Ostrovitianov str. 1, 117997 Moscow, Russia

**Keywords:** advanced structured materials, magnesium fluoride, zinc sulfide, carbon nanotubes, laser-oriented deposition technique

## Abstract

Due to difficulties in obtaining monomaterials, intensive research into the properties of ceramic compositions has been undertaken, along with developments to the properties of the compositions. These are not inferior to monomeric structures for a number of basic parameters. Among the different types of ceramics, magnesium fluoride and zinc sulfide occupy a special place due to their unique properties and specific applications. In this paper, we studied functional optoelectronics and modulating technique elements based on the advanced ceramics MgF_2_ and ZnS. The results of the transmittance spectral parameters and the contact angle estimation as well as an AFM analysis of the studied ceramics, both pure and structured with carbon nanotubes, are presented. We observed that the main characteristics of the studied materials with a surface modified by carbon nanotubes could be significantly changed when an innovative laser-oriented deposition method was applied. This method permitted the CNTs to be deposited in a vertical position on the material surface. The main features of the carbon nanotubes—such as the smaller value of the refractive index, the greater strength between the carbon atoms, and the effective surface—were taken into consideration. The analytical, quantum chemical, and experimental results of the studies of the changes in the basic physical parameters of the selected model of the inorganic matrices of the ceramics are given.

## 1. Introduction

It is well-known that the study of the properties of optoelectronic materials, taking into account the structuring of their volume or the surface, is carried out at an interdisciplinary level and attracts the attention of specialists from different fields of material science. At the same time, it is the process of nanostructuring that contributes to the most detailed manifestation in changing the fundamental properties of the materials.

In the last two decades, the intense creation of the scientific foundations of the nanostructuring process of the materials has promoted the discovery of new physical–chemical properties of the traditional classical compositions, extending the area of their practical use. The interest in nanotechnologies has been caused by the fact that a significant correlation between the spectral, mechanical, refractive, electrochemical, photoconductive, and biocompatible features of the materials has been found, especially in the nanoscale range. Moreover, such experiments are revealing and well-visualized, which allows them to be used in the educational process. It is well-known that a large number of academic teams and engineering communities have investigated the structuring process when the volume or the surface of the inorganic or organic materials have been treated with, for example, halogen elements (F, Br, and I) and carbon nanoparticles, especially CNTs [1,2,3,4,5,6,7,8,9,10,11,12]. The choice of CNTs was conditioned by the unique features of CNTs; namely, their smaller value of refractive index, their greater strength of C–C bonds, and their effective branched surface [13,14,15,16]. Studying the process of ceramic materials allows one to replace conventionally used crystalline bulk systems, the preparation of which is complicated and laborious.

Therefore, it is useful to investigate the process of structuring the surfaces of MgF_2_ and ZnS ceramic materials. The first material is interesting from the point of view that it is, among other things, an exit window for UV lamps that are used in hospitals and polyclinics for the disinfection of premises. The second material is of interest because it is a dividing element for a number of IR lasers. Therefore, high wear resistance is required from these materials.

What are the problems being solved here, but not completely solved? When using magnesium fluoride as a protective element, as a wide-band window in sensors, or as a quarter-wave plate [17,18], it is necessary to obtain a good surface quality. Moreover, it is desirable that this surface serves over time without being subjected to scratches, the penetration of moisture, acids, or other media that can damage the surface and, therefore, affect the basic parameters. Using zinc sulfide, it is important to note that this material is a good modulating layer for electrically- and optically-addressed light modulators, including in circuits for fiber lasers, and can be used for the separation systems of IR lasers [19,20]. Therefore, a good surface is again needed to prevent, for example, parasitic light scattering.

In this work, we present the experimental data of studying the changes in the spectral, mechanical, and wetting properties of the chosen MgF_2_ and ZnS ceramics, obtained by means of an innovative laser-oriented deposition (LOD) technique [21,22] where carbon nanotubes (CNTs) were introduced onto the surface of the materials in a vertical position. The experimental results were confirmed by analytical and quantum chemical calculations.

## 2. Materials and Methods

Due to the fact that laser methods differ significantly from classical CVD and PDV methods by their non-contact effect and by the presence of a lower temperature in the area of the substrates, we preferred to use the laser approach.

A basic laser technique was used in order to create the nanostructuring process. A CO_2_ laser, operated at a wavelength of 10.6 µm with a polarized radiation of 30 W power, was used as the unique approach for the modification of the surface of the studied ceramic materials. Single-walled carbon nanotubes (SWCNTs) with a diameter of 0.4–0.7 nm, type #704121, purchased from Aldrich Co., were used as the nano-objects. They were deposited with a vertical orientation onto the surface of the materials via the application of an electric field of 100–600 V cm^–1^. Part of the setup used for the laser-oriented deposition of the CNTs was given in [23,24]. One observed that a metal grid with a cell size of ~100 μm should be set up close to the substrate with the treated materials. The distance from the grid to the substrate was controlled, which allowed us to vary the strength of the applied electric field used for the orientation of the CNTs in the (mostly) vertical position. 

By using the LOD technique (can be find in Appendix A), we could obtain a few advantages: nanotube-based coatings can withstand flesh heating up to 1500 °C (due to the fact that pure nanotubes themselves are destroyed at a temperature of 2000 °C and above); we did not require the use of toxic solvents; and a non-contact method was applied by the laser deposition of the carbon nanotubes with a diameter of 2–7 nm oriented in an electric field. The thickness of the developed carbon coatings was 10–100 nm; with such a relatively small thickness due to covalent binding to the surface layer of the matrix material, the coatings could withstand mechanical loads from tenths to units and tens of GPa. They also levelled the interface of the media (e.g., material–air and material–vapor water), excluding, in the same way, contact with the atmosphere of the matrix-hardened material. 

The studied materials, e.g., the MgF_2_ materials, were purchased from Electrosteklo LLC. (Saint Petersburg, Moscow, Russia); the ZnS structures were purchased from Alkor Tech Co. (Saint Petersburg, Russia).

The spectra of the CNT-treated ceramics were obtained using Lambda 9 and Furrier FSM-1202 instruments as well as a VIS SF-26 spectrophotometer operated at the range of 250–1200 nm. A Polam-P312 microscope was used to view the image of the treated materials. The microhardness was revealed using a PMT-3M instrument (this device is produced by “LOMO”, Saint Petersburg, Russia) with the ability to vary the indenter forces. An OCA 15EC device was purchased from LabTech Co. (Saint Petersburg, Moscow, Russia) and was used to control the wetting angle (contact angle) change to predict the Lotus effect. The modified surface analysis was formulated using Solver Next AFM (purchased from NT MDT Co., Zelenograd, Moscow, Russia).

As the studied and treated materials, ceramic types MgF_2_ and ZnS were used. The MgF_2_ structure was chosen for this study due to its wide spectral range of operation from 120 nm up to 8 µm and because, it should be repeated, this material is the exit window of the UV lamps that disinfect rooms in hospitals and polyclinics; therefore, it is important to strengthen these exit surfaces. The choice of ZnS was because of the wide use of this material in solar energy systems, semiconducting devices, and the laser schemes of mid-IR range beam splitters; moreover, the lattice parameters of ZnS are close to 0.54 nm, which coincided with the diameters of the CNTs. 

Naturally, many scientific and technical teams investigate both crystalline materials and ceramics using different methods. The manifestation of defects in the studied structures affects the basic parameters [25,26]. However, our non-contact laser method allowed us to organize the binding of the CNTs to the surface layers of the matrix (as the monocrystal or the ceramic), forming covalent bonds, which must be proven in the future by different methods.

Thus, the use of the LOD approach allowed us to predict the formation of the covalent bonding between the CNTs and the surface atoms of the materials. As a result, we could reveal the increase in the transmission, mechanical, and laser strength as well as the change in the wetting angle. 

The quantum chemical calculations were performed by means of the density functional theory (DFT) implemented in the VASP package [27,28] within the augmented plane-wave basis set [29]. The considered systems consisted of 112 Mg atoms and 224 fluorine atoms, 120 magnesium atoms, and 240 fluorine atoms; the CNT consisted of 110 carbon atoms and 10 hydrogen atoms. Due to the large size of the considered system, the plane-wave cutoff energy was set to 250 eV and the density of the states was calculated only at the Г-point. 

## 3. Results and Discussion

In our research, we were able to observe the most significant changes in the properties of the materials under study, taking into account the structuring of their surface.

A few of the data of the changing properties of the chosen ceramics are shown in Table 1 and Table 2 and in Figure 1 and Figure 2.

Table 1 shows the data on the spectral changes of the MgF_2_ and ZnS ceramics. A dramatic increase in the transmittance was observed in the UV range at 125 nm for the MgF_2_ materials. The vacuum UV radiation of excimer lamps was used at wavelengths of 126 nm. An Er^3^+:YAG laser with a pulse width of 500 µs operated in the varied spectral range of 2.3–2.9 µm was applied to study the spectral change, which was measured directly at a wavelength of 2.3 microns via spectrophotometers.

Using randomly chosen samples, the transmittance after the deposition of the CNTs on the surface increased from 5 to 15%. The transmittance of the ZnS materials changed slightly, approximately from 0.5 to 1%. Despite the good coincidence of the diameter of the nanotubes and the lattice parameter of zinc sulfide, a large gap in the values of their refractive index *n* (for ZnS, *n* was close to 2.4 in the VIS spectral range and *n* was close to 2.3 in the near-infrared spectral region). This did not allow us to reveal a sufficient increase in the transmission. On the other hand, the refractive parameters of the CNTs (*n*~1.1) and magnesium fluoride (*n*~1.38) were close to each other, which most likely influenced a significant increase in the transmission. As previously stated, the CNTs had a refractive index close to 1.05–1.1. Our materials had larger refractive index; thus, due to the decrease in the Fresnel losses, the transmittance could be increased. The classical general formula to analytically calculate the Fresnel losses is shown in [30]. It includes the refractive index of the first and second media, which is shown in Equation (1).
(1)$$R=\frac{{\left(n_{2}-n_{1}\right)}^{2}}{{\left(n_{2}+n_{1}\right)}^{2}}$$

Using this formula, one can find a large difference in the spectral parameters regarding reflection and transmittance. However, for the current materials, an additional scattering from the ceramic body could influence the variation in the transmittance and the reflection values.

The data showing the microhardness changes are given in Table 2. The increase in the microhardness for the studied materials was 5–6%. The observed increase in the microhardness was naturally associated with a high Young’s modulus of the CNTs, which also allowed an increase in the strength of the new composites.

The observed tendency of the transmittance and the microhardness changes agreed well with the change in the wetting angle. Thus, the wetting angles changed from 56°–57° to 81°–83° for the MgF_2_ ceramic. The results of the contact angle changes are shown in Figure 1.

The wetting angles changed from 82° to 92° for the ZnS ceramic. The results of the contact angle changes are shown in Figure 2.

To obtain an insight into the influence of the CNT on the physical properties of an initial substrate, we performed ab initio calculations of the density of states in pristine and modified MgF_2_ substrates. The presence of covalent bonding between the CNT and MgF_2_ led to changes to the electronic properties of the latter. The atomic geometry with the main parameters of the considered structure is presented in Figure 3a. Two types of MgF_2_ surfaces were considered; namely, (110) and (111), shown in the top and bottom panels of Figure 3a. Due to the large size of the considered unit cell, the model of the CNT consisted of a relatively small cluster with a dimension of 1.2 nm. The free end on the CNT was passivated by hydrogen. The main changes to the atomic geometry usually directly occurred on the interface; thus, a short nanotube was enough to describe all changes, including the electronic properties. 

In Figure 3b, the dependence of the density of states on the energy of the pristine and the modified surfaces is presented. It was found that the adsorbed CNT led to the formation of additional electronic bands (red curve) within the band gap of the MgF_2_ slab, which is schematically highlighted by the grey rectangle. Such evidence indicates the influence of the CNT adsorbed on the surface on the optical properties. These effects were also observed in a set of semiconducting and metal materials [31,32]. The quantum chemical approach for ZnS materials made in [33] was also supported by the interaction of the carbon nanotubes with the surface atoms of the ZnS matrix and predicted the covalent bonding with them.

Many scientists have used different methods for such simulations—for example, in [34,35]—but we applied the DFT approach in the VASP package presented in [27,28]. It also coincided well with the analytical calculations. 

Analyzing the experimental and modeling results, AFM images were produced. These data are presented in Figure 4.

The analysis was undertaken using five samples, which were randomly chosen. It could be observed that the roughness of the treated magnesium fluoride changed. The basic data are shown in Table 3.

For the ZnS materials, the analogous AFM data were shown in [33]. In the current paper, we partially extended the previous results received for ZnS structures. The change in the microhardness of the monostructure of ZnS (6.5%) and of the ceramic structure ZnS (5%) were revealed and an improvement to the wetting angle of the ceramic matrix was obtained. This coincided with the influence of the defects on this parameter.

## 4. Conclusions

To summarize the obtained results, the following conclusions can be made. 

1. When we applied a laser-oriented deposition technique to the carbon nanotubes placement on the surface of the materials (MgF_2_ and ZnS optical ceramics), it led to noticeable spectral shifts in the UV region for MgF_2_ and slight spectral changes in the mid-IR range for ZnS, along with significant strength changes for both MgF_2_ and ZnS and a change in the wetting angle for these substrates. This last circumstance may directly promote the use of the structured MgF_2_ optical ceramics for improvements of the strength of the UV lamps output window and for orienting a liquid–crystal (LC) molecule instead use of other orienting layers. Moreover, ZnS structured ceramics can reveal two role: as the alignment relief suitable for an orientation of the LC mesophase along one direction in the electrically- or optically-addressed modulators operated in the broad spectral range and as the effective photolayer. 

It should be noted that the production of ceramic materials is much easier than the growth of single crystals; however, it is important to state that the properties of ceramics are not inferior to the similar parameters of the monostructures.

2. Ab initio calculations showed the appearance of additional electronic bands within the band gap of the pristine MgF_2_ slabs. Such results demonstrate the influence of the CNT on the optical properties of the substrate. These results showed the agreement between the experiment and simulation data. 

3. Indeed, the obtained data should be confirmed by other experimental methods in the future; for example, using a mass-spectrometric analysis and Raman spectroscopy. An SEM/TEM instrument would be applied for this aim as well. However, even at this step we can claim that the laser methods had a significant advantage over the conventional PDV and CVC techniques due to contactless and controlled exposure and by a low loss of the substance used for the sedimentation. It can provoke a good reproducibility of the results. 

The results of this work were partially presented at a workshop at Vavilov State Optical Institute (Saint Petersburg) and at the Petersburg Nuclear Physics Institute of the National Research Center, Kurchatov Institute (Gatchina, St. Petersburg, October–November 2017 and 2019), at the Cluster conference in Suzdal (1–6 July 2018), and at the Tambov Graphene conference (6–8 October 2021).

## Figures and Tables

**Figure 1 materials-15-04780-f001:**
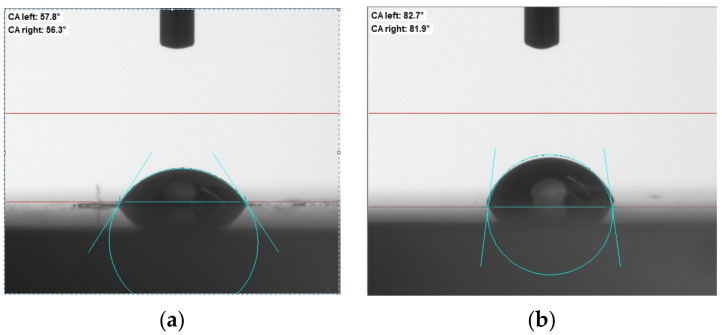
Change in the wetting angle for the pure MgF_2_ (**a**) and for the MgF_2_ + CNTs (**b**).

**Figure 2 materials-15-04780-f002:**
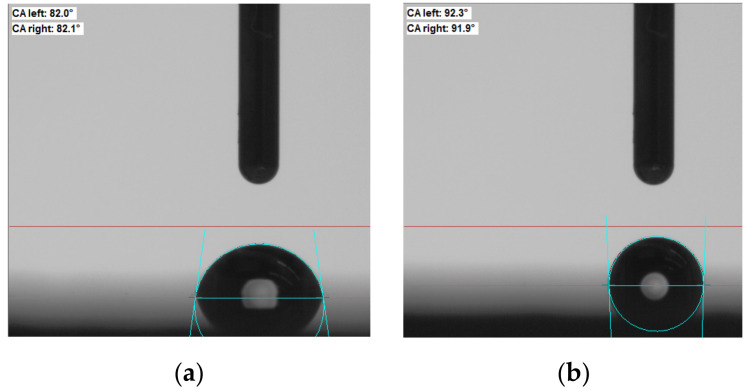
Change in the wetting angle for the pure ZnS (**a**) and for the ZnS + CNTs (**b**).

**Figure 3 materials-15-04780-f003:**
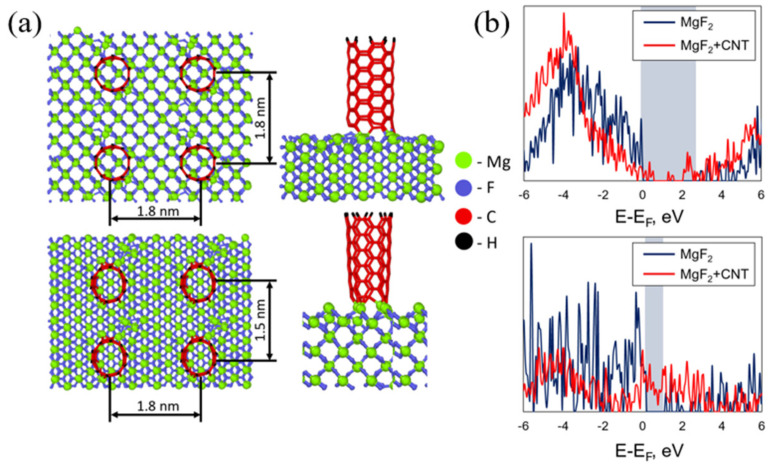
(**a**) Atomic structure of the considered CNT/MgF_2_ interfaces. (**b**) Electronic density of states. Blue and red curves indicate pristine and modified MgF_2_, respectively.

**Figure 4 materials-15-04780-f004:**
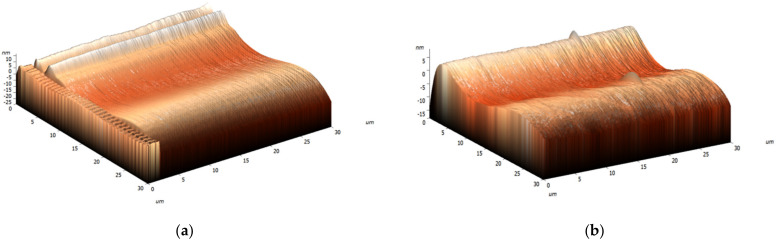
Change in the roughness for the pure MgF_2_ (**a**) and for the MgF_2_ + CNTs (**b**).

**Table 1 materials-15-04780-t001:** Transmittance changes of MgF_2_ and ZnS after the deposition of carbon nanotubes.

Number of Treated Samples ^1^	Transmittance of MgF_2_ at a Wavelength of 125 nm, %		Transmittance of ZnS at a Wavelength of 2.3 μm, %	
	Pure Samples	After CNT Deposition Process	Pure Samples	After CNT Deposition Process
1	61.8	66.6	72	73
2	63.6	69	73	73.5
3	54.5	65.8	71	72.5
4	58.1	67.5	73.5	74
5	50.9	65	73.5	73.8

^1^ The samples were randomly chosen.

**Table 2 materials-15-04780-t002:** Data of the microhardness increase in MgF_2_ and ZnS optical ceramics after the deposition of CNTs.

Ceramic Type	Microhardness Average Value, 10^9^ Pa	Microhardness Increase After Structuring, %	Ceramic Type	Microhardness Average Value, 10^9^ Pa	Microhardness Increase after Structuring, %
MgF_2_	0.3143	~6	ZnS	0.2121	~5
MgF_2_ + CNTs	0.3322		ZnS + CNTs	0.2231	

**Table 3 materials-15-04780-t003:** Result of the roughness value changes for pure MgF_2_ and for MgF_2_ + CNTs.

Area/Parameter	Average Roughness, nm	RMS Roughness, nm	Kurtosis	Skewness
Pure MgF_2_				
1	1.0	1.7	21.175	−2.277
2	3.9	5.3	8.216	−1.349
3	5.1	7.4	9.148	−1.248
4	5.4	7.6	8.247	−1.241
5	1.6	2.5	24.512	−2.875
MgF_2_ + CNTs				
1	2.6	3.3	4.245	−0.586
2	2.3	3.0	6.602	−1.038
3	1.2	1.6	7.769	−1.253
4	2.6	3.3	4.009	−0.743
5	1.5	1.8	3.778	−0.724

## Data Availability

Not applicable.

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
