# Peer review of "Advanced and Functional Structured Ceramics: MgF2 and ZnS"

_materials, 2022, doi:10.3390/ma15144780_

Round 1

Reviewer 1 Report

No characterization like the TEM/SEM, XRD etc are provided. If they are given in earlier paper then this manuscript is just an increment. Language also is turse.  The content of the paper is not enough to warrant a publication. Title also doesn't suggest the content. where is the CNTs, how do you know how the interaction is. 

Author Response

Dear Reviewer!

Thanks a lot for your kind job to see our paper and to make the recommendation for it. Sorry, I have no SEM/TEM instrument in my Lab in order to make the good reply to your comments. But, I have improved a little bit the text body of the paper; all paragraphs added are collared with yellow.

Best Regards,

Natalia Kamanina

=======================================

Natalia V. Kamanina (Prof., Dr.Sci., PhD)

Head of the lab for Photophysics of media with nanoobjects

Vavilov State Optical Institute

Kadetskaya Liniya V.O., dom.5, korpus 2,

St.- Petersburg, 199053, Russia

Professor of the St.-Petersburg Electrotechnical University (“LETI”),

Part-time Leading Researcher at Nuclear Physics Institute (Gatchina)

Job phone: +7 (812) 327-00-95

Fax: +7 (812) 331-75-58 (for N.V.Kamanina)

http://www.photophysics-lab.org/ 

https://publons.com/researcher/1696479/natalia-kamanina/

https://sciprofiles.com/news-feed

http://rusnor.org/network/webinars/10203.htm

http://www.npkgoi.ru/?module=articles&c=profil&b=7

http://www.nanometer.ru/2007/08/09/liquid_crystal_3905.html

http://www.eltech.ru/ru/fakultety/fakultet-elektroniki/sostav-fakulteta/kafedra-kvantovoy-elektroniki-i-optiko-elektronnyh-priborov/sostav-kafedry

=======================================

Comments and Suggestions for Authors

No characterization like the TEM/SEM, XRD etc are provided. If they are given in earlier paper then this manuscript is just an increment. Language also is turse.  The content of the paper is not enough to warrant a publication. Title also doesn't suggest the content. where is the CNTs, how do you know how the interaction is. 

Reviewer 2 Report

This version of the manuscript does not look worthy and cannot be recommended for publication in this form and at least needs a VERY MAJOR REVISION.

1.     Abstract. Line 22.  Please explain and indicate which "spectral parameters".

2.     Introduction.  Motivation of this research is described at a very primitive level. The first paragraph is very general and has nothing to do with this manuscript. After such an introduction, the manuscript could continue to be about any other material (SiO2 or NaCl or any others)  and because of this, nothing had to be changed at all.

3.     The 2nd short paragraph, although it mentions the objects of the study, still does not AT ALL reveal the relevance of the study.  Therefore it is not clear why all these studies were carried out.  Furthermore, these studies have been carried out on ceramics made at home institute, and as we may see, the quality of ceramics and their comparison with other commercial (available to everyone) has not been carried out.

4.     Table 1.  The data given in the table is difficult to follow, as there are no experimental curves. How was the transmission at 125 nm obtained?

Where are the full spectra, where is the comparison with single crystals?

Where does it generally follow that these ceramics are made of MgF2?

5.     The same questions are also about ZnS.

6.     The characterization of ceramics as pure is absolutely incorrect, because such characteristics as defectiveness are omitted. Even in crystals, point defects play a significant, and in many cases simply decisive. See, for example:

Lisitsyn, V. M., et al (2016). Stabilization of primary mobile radiation defects in MgF2 crystals. Nuclear Instruments and Methods in Physics Research Section B: Beam Interactions with Materials and Atoms374, 24-28.

https://doi.org/10.1016/j.nimb.2015.08.002

Ummartyotin, S., & Infahsaeng, Y. (2016). A comprehensive review on ZnS: From synthesis to an approach on solar cell. Renewable and Sustainable Energy Reviews55, 17-24. https://doi.org/10.1016/j.rser.2015.10.120

7.     The data of ab initio calculations of the density of states in the original and modified MgF2 substrate are described too briefly, how the optimization was carried out, what convergence was, what structural data were obtained - is it not clear ! Note, that there several known in literature, different calculations of bulk and surface states of MgF2.

Vassilyeva, A. F., Eglitis, R. I., Kotomin, E. A., & Dauletbekova, A. K. Ab initio calculations of the atomic and electronic structure of MgF2 (011) and (111) surfaces. Central European Journal of Physics9(2), 515-518 (2011).

Usseinov, A. B., Gryaznov, D., et al . Ab initio calculations of pure and Co+ 2-doped MgF2 crystals. Nuclear Instruments and Methods in Physics Research Section B: Beam Interactions with Materials and Atoms470, 10-14 (2020).

Therefore, more detailed presentation and detailing of the obtained calculation data and their comparison with literature and other calculations is absolutely necessary.

8.     It does not at all follow from the work that the authors are working with MgF2 and ZnS ceramics, and in this regard, it is necessary to supplement the work with such supporting data so that such doubts do not arise.

Author Response

Dear Reviewer!

Thanks a lot for your kind job to see our paper and to make the recommendation for it. These recommendations can improve the paper, it is true.

Please see my answers to your questions. The paragraphs added and changed are collared with yellow.

Best Regards,

Natalia Kamanina

Comments and Suggestions for Authors

This version of the manuscript does not look worthy and cannot be recommended for publication in this form and at least needs a VERY MAJOR REVISION.

  1. Abstract. Line 22. Please explain and indicate which "spectral parameters".

In the Abstract section the following is added: “…The results of the transmittance spectral parameters, contact…”

  1. Introduction. Motivation of this research is described at a very primitive level. The first paragraph is very general and has nothing to do with this manuscript. After such an introduction, the manuscript could continue to be about any other material (SiO2 or NaCl or any others)  and because of this, nothing had to be changed at all.

Thank you! I have added the paragraph in the Introduction section: “In this vein, it is useful to investigate the process of structuring the surfaces of magnesium fluoride and zinc sulfide ceramic materials. The first material is interesting from the point of view that it is, among other things, an exit window for UV lamps that are used in hospitals and polyclinics for disinfection of premises. The second material is of interest because it is a dividing element for a number of IR lasers. Therefore, high wear resistance is required from these materials.”

  1. The 2nd short paragraph, although it mentions the objects of the study, still does not AT ALL reveal the relevance of the study. Therefore it is not clear why all these studies were carried out.  Furthermore, these studies have been carried out on ceramics made at home institute, and as we may see, the quality of ceramics and their comparison with other commercial (available to everyone) has not been carried out.

I have added in the Materials section the information about the firms: “Studied materials, such as: MgF2 materials have been purchased from Electrosteklo LLC, (Saint-Petersburg-Moscow, Russia); ZnS structures have been purchased from Alkor Tech Co. (Saint-Petersburg, Russia).”

  1. Table 1. The data given in the table is difficult to follow, as there are no experimental curves. How was the transmission at 125 nm obtained?

Vacuum UV radiation of an excimer lamps has been used at the wavelengths of 126 nm.

Where are the full spectra, where is the comparison with single crystals?

Where does it generally follow that these ceramics are made of MgF2?

Sorry, I have no spectral curves, but the values of the spectral parameters at the unique wavelength are true.

  1. The same questions are also about ZnS.

Er3+:YAG-laser with the pulse width of 500 ms operated in the varied spectral range of 2,3-2,9 micrometers has been applied to study the spectral change, which has been measured directly at the wavelength of 2,3 microns via spectrophotometers.

Sorry, I have no spectral curves, but the values of the spectral parameters at the unique wavelength are true.

  1. The characterization of ceramics as pure is absolutely incorrect, because such characteristics as defectiveness are omitted. Even in crystals, point defects play a significant, and in many cases simply decisive. See, for example:

Lisitsyn, V. M., et al (2016). Stabilization of primary mobile radiation defects in MgF2 crystals. Nuclear Instruments and Methods in Physics Research Section B: Beam Interactions with Materials and Atoms374, 24-28.

https://doi.org/10.1016/j.nimb.2015.08.002

Ummartyotin, S., & Infahsaeng, Y. (2016). A comprehensive review on ZnS: From synthesis to an approach on solar cell. Renewable and Sustainable Energy Reviews55, 17-24. https://doi.org/10.1016/j.rser.2015.10.120

I have added your references for the recommendation. They are now as [21,22].

Naturally, so many scientific and technical teams investigate both crystalline materials and ceramics by different methods. Indeed, the manifestation of the defects in the studied structures affect the basic parameters [21,22]. But, our non-contact laser method allows us to organize the binding of CNTs to the surface layers of the matrix (it can be as the mono crystals or the ceramics one), forming covalent bonds, which, of course, must be proved in the future by different methods.

  1. The data of ab initio calculations of the density of states in the original and modified MgF2 substrate are described too briefly, how the optimization was carried out, what convergence was, what structural data were obtained - is it not clear ! Note, that there several known in literature, different calculations of bulk and surface states of MgF2.

Vassilyeva, A. F., Eglitis, R. I., Kotomin, E. A., & Dauletbekova, A. K. Ab initio calculations of the atomic and electronic structure of MgF2 (011) and (111) surfaces. Central European Journal of Physics9(2), 515-518 (2011).

Usseinov, A. B., Gryaznov, D., et al . Ab initio calculations of pure and Co+ 2-doped MgF2 crystals. Nuclear Instruments and Methods in Physics Research Section B: Beam Interactions with Materials and Atoms470, 10-14 (2020).

Therefore, more detailed presentation and detailing of the obtained calculation data and their comparison with literature and other calculations is absolutely necessary.

I have added some paragraph using your recommended papers, they are [29,30]:

It should be mentioned that, indeed, many scientists have used different methods for the simulations, for example, shown in [29,30], but we have applied DFT approach in the VASP package presented in [23,24]. It can be in good coinciding with the analytical calculation as well.

  1. It does not at all follow from the work that the authors are working with MgF2 and ZnS ceramics, and in this regard, it is necessary to supplement the work with such supporting data so that such doubts do not arise.

Sorry, I don't have SEM/TEM instruments in my laboratory, I will plan to do such experiments in the future.

Thank you once again for your useful job for us, which really cam improve our paper.

Best Regards,

Natalia Kamanina

=======================================

Natalia V. Kamanina (Prof., Dr.Sci., PhD)

Head of the lab for Photophysics of media with nanoobjects

Vavilov State Optical Institute

Kadetskaya Liniya V.O., dom.5, korpus 2,

St.- Petersburg, 199053, Russia

Professor of the St.-Petersburg Electrotechnical University (“LETI”),

Part-time Leading Researcher at Nuclear Physics Institute (Gatchina)

Job phone: +7 (812) 327-00-95

Fax: +7 (812) 331-75-58 (for N.V.Kamanina)

http://www.photophysics-lab.org/ 

https://publons.com/researcher/1696479/natalia-kamanina/

https://sciprofiles.com/news-feed

http://rusnor.org/network/webinars/10203.htm

http://www.npkgoi.ru/?module=articles&c=profil&b=7

http://www.nanometer.ru/2007/08/09/liquid_crystal_3905.html

http://www.eltech.ru/ru/fakultety/fakultet-elektroniki/sostav-fakulteta/kafedra-kvantovoy-elektroniki-i-optiko-elektronnyh-priborov/sostav-kafedry

=======================================

Reviewer 3 Report

The authors presented the results on some characteristics of MgF2 and ZnS ceramics modified by CNTs. In general, the subject  of the submitted manuscript fully corresponds to the goals of the journal "Materials". However, the article cannot be published in its present form. Below are my comments.

(1) Since CNTs receive a lot of attention in the manuscript, I recommend expanding the list of keywords by adding words related to carbon nanotubes.

(2) The "Introduction" section should briefly describe the current state of research on MgF2 and ZnS ceramics, including research on carbon nanotubes. What has been done in this field, why was the laser-oriented deposition method chosen for the synthesis? Why were these materials chosen for the study? These points are partially covered in the section “Materials and methods”. Thus, I recommend reorganizing the two sections in order to enhance the Introduction section and emphasize the significance of the research presented.

(3) Lines 168 - 171. "The change in the microhardness of the monostructure of ZnS (6,5%) ...can be coincided with the defects influence on this parameter".  - This block should be moved to the section Introduction as well.

(4) Line 167. "In the current paper we have partially extended the results received for the ZnS structures before" - Why was it necessary to extend the results previously obtained for ZnS ceramics?

(5) How can the authors explain such a significant difference in the transmittance  change  for MgF2 and ZnS ceramics modified by CNTs?

(6) From my point of view, the work is more descriptive, and I recommend adding some critical remarks for the benefit of readers.

Author Response

Dear Reviewer!

Thanks a lot for your kind job to see our paper and to make the recommendation for it. These recommendations can improve the paper, it is true.

Please see my answers to your questions. The paragraphs added and changed are collared with yellow.

Best Regards,

Natalia Kamanina

Comments and Suggestions for Authors

The authors presented the results on some characteristics of MgF2 and ZnS ceramics modified by CNTs. In general, the subject  of the submitted manuscript fully corresponds to the goals of the journal "Materials". However, the article cannot be published in its present form. Below are my comments.

(1) Since CNTs receive a lot of attention in the manuscript, I recommend expanding the list of keywords by adding words related to carbon nanotubes.

Thank you! I have added the word: carbon nanotubes in the keywords list.

(2) The "Introduction" section should briefly describe the current state of research on MgF2 and ZnS ceramics, including research on carbon nanotubes. What has been done in this field, why was the laser-oriented deposition method chosen for the synthesis? Why were these materials chosen for the study? These points are partially covered in the section “Materials and methods”. Thus, I recommend reorganizing the two sections in order to enhance the Introduction section and emphasize the significance of the research presented.

Thank you! I have added in the Introduction part the following: In this vein, it is useful to investigate the process of structuring the surfaces of magnesium fluoride and zinc sulfide ceramic materials. The first material is interesting from the point of view that it is, among other things, an exit window for UV lamps that are used in hospitals and polyclinics for disinfection of premises. The second material is of interest because it is a dividing element for a number of IR lasers. Therefore, high wear resistance is required from these materials.

In the Materials part I have added: It should be mentioned that using LOD technique we can obtain some advantages: nanotube-based coatings can withstand flesh heating up to 1500 °C (due to the fact that pure nanotubes themselves are destroyed at a temperature of 2000 °C and above); do not require the use of toxic solvents; are applied by a non-contact method by the laser deposition of the carbon nanotubes with a diameter of 2-7 nm oriented in an electric field. The thickness of the developed carbon coatings is 10-100 nm; with such a relatively small thickness, due to covalent binding to the surface layer of the matrix material, the coatings withstand mechanical loads – from tenths to units and tens of GPa; they also level the interface of media: material-air, material-vapor water, excluding, in the same way, contact with the atmosphere of the matrix hardened material.

(3) Lines 168 - 171. "The change in the microhardness of the monostructure of ZnS (6,5%) ...can be coincided with the defects influence on this parameter".  - This block should be moved to the section Introduction as well.

Sorry, I have save (keep) this sentence in this place.

(4) Line 167. "In the current paper we have partially extended the results received for the ZnS structures before" - Why was it necessary to extend the results previously obtained for ZnS ceramics?

It can be necessary because before the mono-structure has been studied, but now we have used the poly-structures.

(5) How can the authors explain such a significant difference in the transmittance  change  for MgF2 and ZnS ceramics modified by CNTs?

Good question!!! Yes, I can explain it. The CNTs have the refractive index close to 1,05-1,1. Our materials have larger refractive index, thus, due to the decrease the Fresnel losses the transmittance can be increased!

(6) From my point of view, the work is more descriptive, and I recommend adding some critical remarks for the benefit of readers.

Yes, it is true. But, sorry, I have now SEM/TEM instruments in my Lab, which can critically show the development of the novel coatings. But, all results are obtained namely in my Lab, using our Lab devices, it is important! Maybe in future I will find the SEM/TEM equipment in other Institute in Russia and make the experiments needed to support carefully the novel coatings.

Thank you once again for your useful job for us, which really cam improve our paper.

Best Regards,

Natalia Kamanina

=======================================

Natalia V. Kamanina (Prof., Dr.Sci., PhD)

Head of the lab for Photophysics of media with nanoobjects

Vavilov State Optical Institute

Kadetskaya Liniya V.O., dom.5, korpus 2,

St.- Petersburg, 199053, Russia

Professor of the St.-Petersburg Electrotechnical University (“LETI”),

Part-time Leading Researcher at Nuclear Physics Institute (Gatchina)

Job phone: +7 (812) 327-00-95

Fax: +7 (812) 331-75-58 (for N.V.Kamanina)

http://www.photophysics-lab.org/ 

https://publons.com/researcher/1696479/natalia-kamanina/

https://sciprofiles.com/news-feed

http://rusnor.org/network/webinars/10203.htm

http://www.npkgoi.ru/?module=articles&c=profil&b=7

http://www.nanometer.ru/2007/08/09/liquid_crystal_3905.html

http://www.eltech.ru/ru/fakultety/fakultet-elektroniki/sostav-fakulteta/kafedra-kvantovoy-elektroniki-i-optiko-elektronnyh-priborov/sostav-kafedry

=======================================

Round 2

Reviewer 1 Report

Claiming nanostructure, CNTs etc and not showing the SEM and TEM results are not acceptable. You can collaborate and provide the results. 

Author Response

Dear Reviewer!

Thank you for your kind response regarding our paper.

Thanks a lot for your job and your recommendations, which indeed should be taken into account.

Comments and Suggestions for Authors

Claiming nanostructure, CNTs etc and not showing the SEM and TEM results are not acceptable. You can collaborate and provide the results. 

So sorry once again, for the period of 10 days, which I have received from the MDPI system, I have no ability to find the SEM or TEM instruments. Now the summer period, thus so many scientists from the Saint-Petersburg Universities have the vacations. If I will ask some my colleagues from the Research Institute it should be also can be correlated with their paln of the research. It is not in my rules to interfere with specialists and ask them to do work for me in a short time. I also help everyone if they ask, but it's better to negotiate in advance. I included a note at the end of the article that SEM, TEM devices are necessary for further work, following your recommendation.

Moreover, all changes now are collared with green.

Best Regards,

Natalia Kamanina

Reviewer 2 Report

The authors have significantly improved the manuscript, taking into account all the comments and recommendations of the reviewer. The article can now be recommended for publication without any doubt.

Author Response

Dear Reviewer!

Thank you for your kind response regarding this paper.

Thanks a lot for your job and your recommendations, which permit to improve our paper.

Comments and Suggestions for Authors

The authors have significantly improved the manuscript, taking into account all the comments and recommendations of the reviewer. The article can now be recommended for publication without any doubt.

Best Regards and be healthy!

Natalia Kamanina

=======================================

Natalia V. Kamanina (Prof., Dr.Sci., PhD)

Head of the lab for Photophysics of media with nanoobjects

Vavilov State Optical Institute

Kadetskaya Liniya V.O., dom.5, korpus 2,

St.- Petersburg, 199053, Russia

Professor of the St.-Petersburg Electrotechnical University (“LETI”),

Part-time Leading Researcher at Nuclear Physics Institute (Gatchina)

Job phone: +7 (812) 327-00-95

Fax: +7 (812) 331-75-58 (for N.V.Kamanina)

http://www.photophysics-lab.org/ 

https://publons.com/researcher/1696479/natalia-kamanina/

https://sciprofiles.com/news-feed

http://rusnor.org/network/webinars/10203.htm

http://www.npkgoi.ru/?module=articles&c=profil&b=7

http://www.nanometer.ru/2007/08/09/liquid_crystal_3905.html

http://www.eltech.ru/ru/fakultety/fakultet-elektroniki/sostav-fakulteta/kafedra-kvantovoy-elektroniki-i-optiko-elektronnyh-priborov/sostav-kafedry

=======================================

Reviewer 3 Report

In my opinion, the submitted manuscript can be published in the journal after serious revision, because it does not correspond to the level of articles published in the journal “Materials”.

According to the instructions to authors on the journal website “The introduction should briefly place … the current state of the research field should be reviewed carefully and key publications cited. Please highlight controversial and diverging hypotheses when necessary. Keep the introduction comprehensible to scientists working outside the topic of the paper”. I come back to the question, has there been any research done in this field? If so, which ones? According to the authors of the article there have been, at least on ZnS by the authors themselves. What problems did previous studies reveal, and why were additional experiments needed?

Lines 50-55. It it necessary to provide corresponding references.

Ceramic samples before and after modification by CNTs should be characterized at least minimally, for example using SEM and XRD techniques.

Author Response

Dear Reviewer!

Thanks a lot for your useful opinion! Some paragraphs added are collared by green.

Comments and Suggestions for Authors

 In my opinion, the submitted manuscript can be published in the journal after serious revision, because it does not correspond to the level of articles published in the journal “Materials”.

According to the instructions to authors on the journal website “The introduction should briefly place … the current state of the research field should be reviewed carefully and key publications cited. Please highlight controversial and diverging hypotheses when necessary. Keep the introduction comprehensible to scientists working outside the topic of the paper”. I come back to the question, has there been any research done in this field? If so, which ones? According to the authors of the article there have been, at least on ZnS by the authors themselves. What problems did previous studies reveal, and why were additional experiments needed?

Lines 50-55. It it necessary to provide corresponding references.

Ceramic samples before and after modification by CNTs should be characterized at least

I have included the references:

Rezende, G.C.; Calve, S. Le., Brandner, J.J. and Newport, D. Micro photoionization detectors. Sensors and Actuators B: Chemical, 2019, Vol. 287, p. 86-94. DOI: 10.1016/j.snb.2019.01.072

Du, Y.; Chen, B.S.; Lin, J.J.; Tseng, H.W.; Wu, Y.L. and Yang, C.F. Fabrication of 500 nm distributed Bragg reflector using Nb2O5-MgF2 multi-layer films. Modern Physics Letters B, 2021, Vol. 35, No. 2140001, p.1-10.  DOI: 10.1142/S0217984921400017

Li, J.; Li, X.; Feng, J.; Zheng, C.; Wang, Y.; Wang; A. and Liu; X. ZnS nanospheres for optical modulator in an Erbium-Doped Fiber Laser. Annalen Der Physik, 2020, Vol. 532(3), 1900454, p. 1-8. DOI: 10.1002/andp.201900454

Nagl, N.; Mak, K.F.; Wang, Q.; Pervak, V.; Krausz ,F. and Pronin, O. Efficient femtosecond mid-infrared generation based on a Cr:ZnS oscillator and step-index fluoride fibers. Optics Letters, 2019, Vol. 44(10), p. 2390-2393. DOI: 10.1364/OL.44.002390

In order to more clear identify the problem and the solution of the desired task.

Moreover, I have included the Fresnel formula for the losses in the Results and Discussion section.

Let us to indicate that the classical general formula to analytically calculate the Fresnel losses is shown in book [30]. It includes the refractive index of the first and second media, which is shown in equation (1).

                                                         (1)

Indeed, using this formula one can find more large difference in the spectral parameters regarding the reflection and transmittance one. But, for the current materials an additional scattering from the ceramic body can influence on the variation of the transmittance and reflections values.

I hope that this variant of the paper can be clearer understudied.

Best Regards, Natalia Kamanina
